# BLOCKDANCE: REUSE STRUCTURALLY SIMILAR SPATIO-TEMPORAL FEATURES TO ACCELERATE DIFFUSION TRANSFORMERS

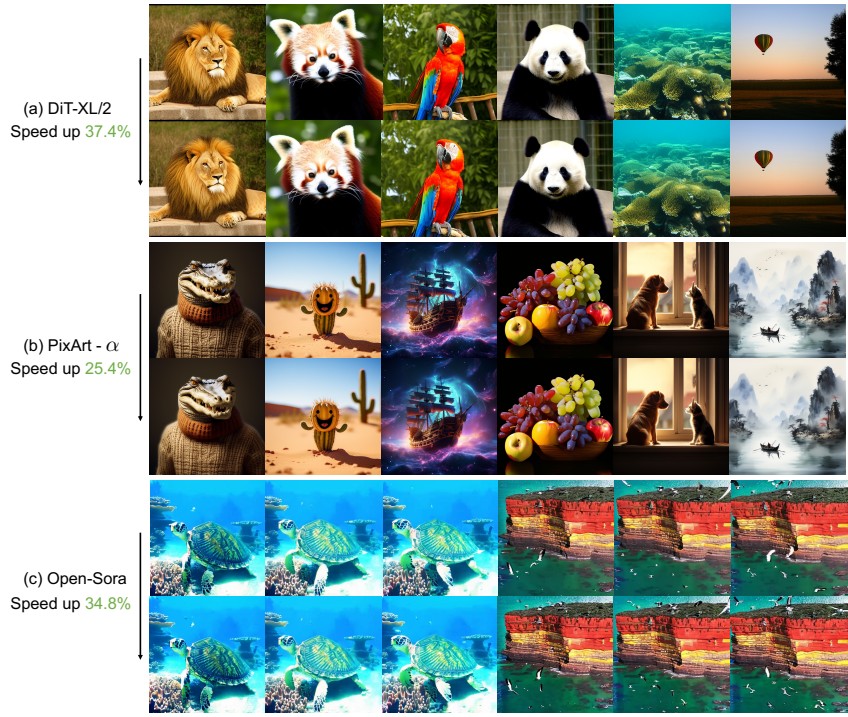

Figure 1: BlockDance accelerates DiT models DiT-XL/2, PixArt-$\alpha$ and Open-Sora by 37.4%, 25.4% and 34.8% respectively, while maintaining fidelity and high consistency with the original image.

## ABSTRACT

Diffusion models have demonstrated impressive generation capabilities, particularly with recent advancements leveraging transformer architectures to improve both visual and artistic quality. However, Diffusion Transformers (DiTs) continue to encounter challenges related to low inference speed, primarily due to the iterative denoising process. To address this issue, we propose BlockDance, a training-free approach that explores feature similarities at adjacent time steps to accelerate DiTs. Unlike previous feature-reuse methods that lack tailored reuse strategies for features at different scales, BlockDance prioritizes the identification of the most structurally similar features, referred to as Structurally Similar Spatio-Temporal (STSS) features. These features are primarily located within the structure-focused blocks of the transformer during the later stages of denoising. BlockDance caches and reuses these highly similar features to mitigate redundant computation, thereby accelerating DiTs while maximizing consistency with the generated results of the original model. Furthermore, considering the diversity of generated content and the varying distributions of redundant features, we introduce BlockDance-Ada, a lightweight decision-making network tailored for instance-specific acceleration. BlockDance-Ada dynamically allocates resources and provides superior content

quality. Both BlockDance and BlockDance-Ada have demonstrated effectiveness across diverse generation tasks and models, achieving an acceleration ranging from 25% to 50% while preserving generation quality.

# 1 INTRODUCTION

Diffusion models have been recognized as a pivotal advancement for both image and video generation tasks due to their impressive capabilities. Recently, there has been a growing interest in shifting the architecture of diffusion models from U-Net to transformers (OpenAI, 2024; Labs, 2024; Zhou et al., 2024). This refined architecture empowers these models not just to generate visually convincing and artistically compelling images and videos, but also to better adhere to scaling laws.

Despite the remarkable performance of these transformer-based diffusion models, their applicability to real-time scenarios remains constrained by slow inference speed, primarily due to the iterative nature of the denoising process. Existing acceleration approaches primarily focus on two paradigms: I) reducing the number of sampling steps through novel scheduler designs (Song et al., 2021; Lu et al., 2022) or step distillation (Ren et al., 2024; Lin et al., 2024); II) minimizing computational overhead per step through the employment of model pruning (Fang et al., 2023; Kim et al., 2023), model distillation (Gupta et al., 2024; Zhang et al., 2024a), or the mitigation of redundant calculations (Wimbauer et al., 2024; Ma et al., 2024b). This paper aims to accelerate DiTs by mitigating redundant computation, as this paradigm can be plug-and-play into various models and tasks. Although feature redundancy is widely recognized in visual tasks (He et al., 2022; Meng et al., 2022), and recent works have identified its presence in the denoising process of diffusion models (Ma et al., 2024b; Li et al., 2023), the issue of feature redundancy within DiT models—and the potential strategies to mitigate these redundant computation—remains obscured from view.

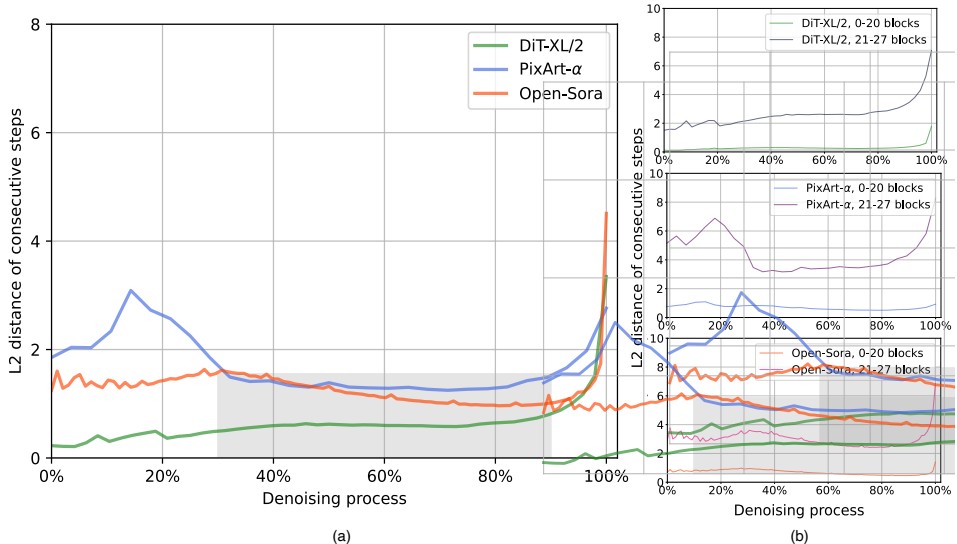

(a)    (b)

Figure 2: **Feature similarity and redundancy in DiTs**. (a) In the denoising process, the outputs of DiT blocks exhibit high similarity in adjacent steps, particularly in the gray shadow-masked region where the structure is stabilized. (b) This high similarity is mainly concentrated in the shallow and middle blocks within the transformer, *i.e.* between 0 and 20 blocks, which focus on low-level structures. Thus, redundant computation related to highly similar structural features in the denoising process can be saved by reusing them to accelerate DiTs inference while maintaining quality.

To this end, we revisit the inter-feature distances between the blocks of DiTs at adjacent time steps in Figure 2 (a) and propose BlockDance, a training-free acceleration approach by caching and reusing highly similar features to reduce redundant computation. Previous feature reuse methods lack tailor-made reuse strategies for features at different scales with varying levels of similarity. Consequently, the reused set often includes low-similarity features, leading to structural distortions in the image and misalignment with the prompt. In contrast, BlockDance enhances the reuse strategy and focuses on

the most similar features, *i.e.* Structurally Similar Spatio-Temporal (STSS) features. To be specific, during the denoising process, structural content is typically generated in the initial steps when noise levels are high, whereas texture and detail content are frequently generated in subsequent steps characterized by lower noise levels (Ho et al., 2020; Hertz et al., 2023). Thus, we hypothesize that once the structure is stabilized, the structural features will undergo minimal changes. To validate this hypothesis, we decouple the features of DiTs at different scales, as illustrated in Figure 2 (b). The observation reveals that the shallow and middle blocks, which concentrate on coarse-grained structural content, exhibit minimal variation across adjacent steps. In contrast, the deep blocks, which prioritize fine-grained textures and complex patterns, demonstrate more noticeable variations. Thus, we argue that allocating computational resources to regenerate these structural features yields marginal benefits while incurring significant costs. To address this issue, we propose a strategy of caching and reusing highly similar structural features subsequent to the stabilization of the structure to accelerate DiTs while maximizing consistency with the generated results of the original model.

Considering the diverse nature of generated content and their varying distributions of redundant features, we introduce BlockDance-Ada, a lightweight decision-making network tailored for Block-Dance. In simpler content with a limited number of objects, we observe a higher presence of redundant features. Therefore, frequent feature reuse in such scenarios is adequate to achieve satisfactory results while offering increased acceleration benefits. Conversely, in the context of intricate compositions characterized by numerous objects and complex interrelations, there are fewer high-similarity features available for reuse. Learning this adaptive strategy is a non-trivial task, as it involves non-differentiable decision-making processes. Thus, BlockDance-Ada is built upon a reinforcement learning framework. BlockDance-Ada utilizes policy gradient methods to drive a strategy for caching and reusing features based on the prompt and intermediate latent, maximizing a carefully designed reward function that encourages minimizing computation while maintaining quality. Accordingly, BlockDance-Ada is capable of adaptively allocating resources.

The main contributions of this paper are summarized in the following:

- We introduce BlockDance, a novel, training-free, and efficient algorithm for accelerating Diffusion Transformers (DiTs). This algorithm caches and reuses Structurally Similar Spatio-Temporal (STSS) features to minimize redundant computation during inference, offering compatibility with various models through a plug-and-play approach.

- Regarding the varying complexity of generated content, we investigate the feasibility of instance-specific BlockDance strategies, and propose BlockDance-Ada, a method that adaptively saves computation through employing reinforcement learning techniques.

- BlockDance has been validated across diverse datasets, including ImageNet, COCO2017, and MSR-VTT. It has been tested in tasks such as class-conditioned generation, text-to-image, and text-to-video using models such as DiT-XL/2, Pixart-$\alpha$, and Open-Sora. Our experimental results reveal that our method can achieve a 25%-50% acceleration in inference speed with training-free while maintaining comparable generated quality. Furthermore, the proposed BlockDance-Ada generates higher-quality content at the same acceleration ratio.

## 2 RELATED WORK

**Diffusion transformer.** Diffusion models (Ho et al., 2020; Dhariwal & Nichol, 2021; Song & Ermon, 2020) have emerged as key players in the field of generation due to their impressive capabilities. Previously, U-Net-based (Ronneberger et al., 2015) diffusion models have demonstrated remarkable performance across various applications, including image generation (Rombach et al., 2022; Podell et al., 2024) and video generation (Wu et al., 2023; Singer et al., 2023; Blattmann et al., 2023). Recently, some research (Peebles & Xie, 2023; Chen et al., 2024a; Zheng, 2024; Ma et al., 2024a; Li et al., 2024; Labs, 2024; Zhou et al., 2024) has transitioned to transformer-based (Vaswani et al., 2017) architectures, *i.e.* Diffusion Transformers (DiTs). This framework excels in generating visually convincing and artistically compelling content, better adheres to scaling laws, and shows promise in efficiently integrating and generating multi-modality content. However, DiT models are still hindered by the inherent iterative nature of the diffusion process, limiting their real-time applications.

**Acceleration of diffusion models.** Efforts have been made to accelerate the inference process of diffusion models, which can be summarized into two paradigms: reducing the number of sampling

steps and reducing the computation per step. The first paradigm often involves designing faster samplers (Song et al., 2021; Lu et al., 2022; Zhao et al., 2023) or step distillation (Meng et al., 2023; Luo et al., 2023; Sauer et al., 2023; Lin et al., 2024; Sauer et al., 2024; Ren et al., 2024). The second paradigm focuses on model-level distillation (Gupta et al., 2024; Zhang et al., 2024a), pruning (Kim et al., 2023; Fang et al., 2023), or reducing redundant computation (Ma et al., 2024b; Bolya & Hoffman, 2023; Li et al., 2023; Wimbauer et al., 2024; So et al., 2024; Zhang et al., 2024b). Several studies (Ma et al., 2024b; Li et al., 2023) have unearthed the existence of redundant features in U-Net-based diffusion models, but their coarse-grained feature reuse strategies include those low-similarity features, leading to structural distortions and text-image misalignment. In contrast, we investigate the feature redundancy in DiTs and propose reusing structurally similar spatio-temporal features to achieve acceleration while maintaining high consistency with the base model's results.

**Reinforcement learning in diffusion models.** Efforts have been dedicated to fine-tuning diffusion models using reinforcement learning (Sutton & Barto, 2018) to align their outputs with human preferences or meticulously crafted reward functions. Typically, these models (Xu et al., 2023; Black et al., 2023; Fan et al., 2023; Lee et al., 2023; Prabhudesai et al., 2023; Kirstain et al., 2023) aim to improve the prompt alignment and visual aesthetics of the generated content. This paper explores learning instance-specific acceleration strategies through reinforcement learning.

## 3 METHOD

### 3.1 PRELIMINARIES

**Forward and reverse process in diffusion modes** Diffusion models gradually add noise to the data and then learn to reverse this process to generate the desired noise-free data from noise. In this paper, we focus on the formulation introduced by (Rombach et al., 2022) that performs noise addition and denoising in latent space. In the forward process, the posterior probability of the noisy latent $\mathbf{z}_t$ at time step $t$ has a closed form:

$$q(\mathbf{z}_t|\mathbf{z}_0) = \mathcal{N}(\mathbf{z}_t; \sqrt{\bar{\alpha}_t}\mathbf{z}_0, (1 - \bar{\alpha}_t)\mathbf{I}), \tag{1}$$

where $\bar{\alpha}_t = \prod_{i=0}^{t} \alpha_i = \prod_{i=0}^{t}(1 - \beta_i)$ and $\beta_i \in (0, 1)$ represents the noise variance schedule. The inference process, *i.e.* the reverse process of generating data from noise, is a crucial part of the diffusion model framework. Once the diffusion model $\epsilon_\theta(\mathbf{z}_t, t)$ is trained, during the reverse process, traditional sampler DDPM (Ho et al., 2020) denoise $\mathbf{z}_T \sim \mathcal{N}(\mathbf{0}, \mathbf{I})$ step by step for a total of $T$ steps. One can also use a faster sampler like DDIM (Song et al., 2021) to speed up the sampling process via the following process:

$$\mathbf{z}_{t-1} = \sqrt{\alpha_{t-1}}\left(\frac{\mathbf{z}_t - \sqrt{1 - \alpha_t}\epsilon_\theta(\mathbf{z}_t, t)}{\sqrt{\alpha_t}}\right) + \sqrt{1 - \alpha_{t-1} - \sigma_t^2} \cdot \epsilon_\theta(\mathbf{z}_t, t) + \sigma_t\epsilon_t. \tag{2}$$

In the denoising process, the model primarily generates rough structures of the image in the early stages and gradually refines it by adding textures and detailed information in later stages.

**Features in the transformer.** The number of denoising steps is related to the number of network inferences in the DiT architecture, which typically features multiple blocks stacked together. Each block sequentially computes its output based on the input from the previous block. The shallow blocks, closer to the input, are inclined to capture the global structures and rough outlines of the data. In contrast, the deep blocks, closer to the output, gradually refine specific details to achieve outputs that are both realistic and visually appealing (Wu et al., 2021; Park & Kim, 2022; Raghu et al., 2021).

### 3.2 FEATURE SIMILARITY AND REDUNDANCY IN DITS

The inference speed of DiTs is constrained by its inherently iterative nature of inference, limiting its practical applicability. This paper aims to reduce redundant computation to accelerate DiTs.

Upon revisiting the denoising process in various DiT models, including DiT-XL/2 (Peebles & Xie, 2023), PixArt-$\alpha$ (Chen et al., 2024a), and Open-Sora (Zheng, 2024), two key findings emerged: I) There are significant feature similarities between consecutive steps, indicating redundant computation in the denoising process, as illustrated in Figure 2 (a); II) This high similarity is primarily manifested

in the shallow and middle blocks (between 0 and 20 blocks) of the transformer, while deeper blocks (between 21 and 27 blocks) exhibit more variations, as depicted in Figure 2 (b). We attribute this phenomenon to the fact that structural content is generally produced in the initial steps, while textures and details are generated in the later steps.

To confirm this, we visualize the block features of PixArt-$\alpha$ using Principal Components Analysis (PCA), as shown in Figure 3. At the initial stages of denoising, the network primarily focuses on generating structural content, such as human poses and other basic forms. As the denoising process progresses, the shallow and middle blocks of the network still concentrate on generating low-frequency structural content, while the deeper blocks shift their focus towards generating more complex high-frequency texture information, such as clouds and crowds within depth of field. Consequently, after the structure is established, the feature maps highlighted by blue boxes in Figure 3 exhibit high consistency across adjacent steps. We define these computation as redundant computation, which relate to the low-level structures that the shallow and middle blocks of the transformer focus on. Based on these observations, we argue that allocating substantial computational resources to regenerate these similar features yields marginal benefits but leads to higher computational costs. Thus, our goal is to design a strategy that leverages these highly similar features to reduce redundant computation and accelerate the denoising process.

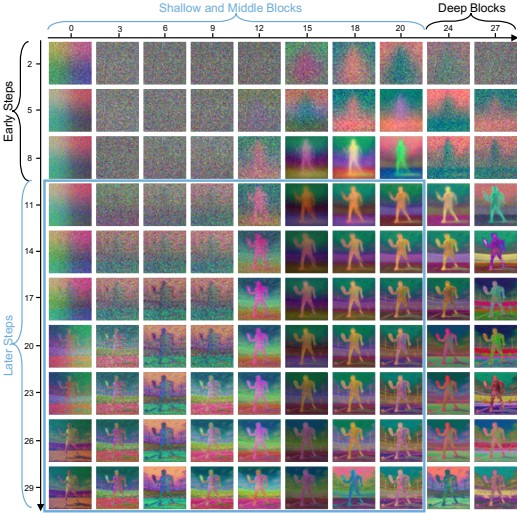
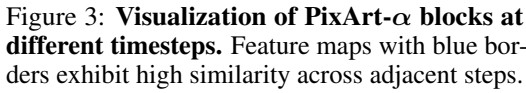

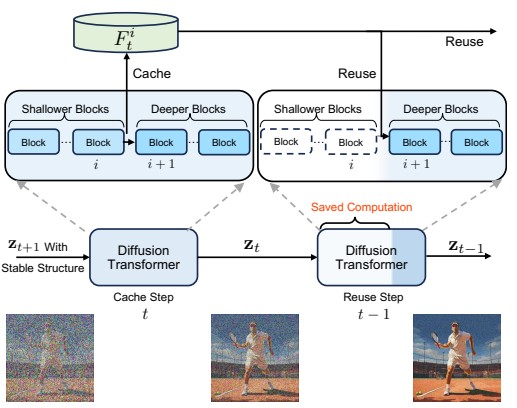

Figure 3: **Visualization of PixArt-$\alpha$ blocks at different timesteps.** Feature maps with blue borders exhibit high similarity across adjacent steps.

Figure 4: **An overview of BlockDance.** The reuse step generates $\mathbf{z}_{t-1}$ based on reusing the structural features from the cache step, thereby saving the computation of the first $i$ blocks to accelerate the inference.

### 3.3 TRAINING-FREE ACCELERATION APPROACH

We introduce BlockDance, a straightforward yet effective method to accelerate DiTs by leveraging feature similarities between steps in the denoising process. By strategically caching the highly similar structural features and reusing them in subsequent steps, we reduce redundant computation.

Specifically, we design the denoising steps into two types: cache step and reuse step, as illustrated in Figure 4. During consecutive time steps, a cache step first conducts a standard network forward based on $\mathbf{z}_{t+1}$, outputs $\mathbf{z}_t$, and saves the features $F_t^i$ of the $i$-th block. For the following time step—the reuse step—we do not perform full network forward computation; instead, we carry out partial inference. More specifically, we reuse the cached features $F_t^i$ from the cache step as the input for the $(i+1)$-th block in the reuse step. Therefore, the computation of the first $i$ blocks in the reuse step can be saved due to the sequential inference characteristic of the transformer blocks, and only the blocks deeper than $i$ require recalculation.

To this end, it is crucial to determine the optimal block index and the stage of the denoising process where reuse should be concentrated. Based on the insights from Figure 2 and Figure 3, we set the index as 20 and focus the reuse on the latter 60% of the denoising process, after the structure has stabilized. These settings enable the decoupling of feature reuse and specifically reuse the structurally similar spatio-temporal features. Thus, we set the first 40% of denoising steps as cache steps and

evenly divide the remaining 60% of denoising steps into several groups, each comprising $N$ steps. The first step of each group is designated as a cache step, while the subsequent $N-1$ steps are reuse steps to accelerate inference. With the arrival of a new group, a new cache step updates the cached features, which are then utilized for the reuse steps within that group. This process is repeated until the denoising process concludes. A larger $N$ represents a higher reuse frequency. We term this cache and reuse strategy as BlockDance-$N$, which operates in a training-free paradigm and can effectively accelerate multiple types of DiT models while maintaining the quality of the generated content.

### 3.4 INSTANCE-SPECIFIC ACCELERATION APPROACH

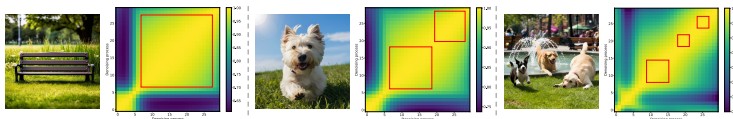

Figure 5: Cosine similarity of the generated images' features during denoising. High similarity step features, highlighted by the red box, decrease as structural complexity increases.

However, the generated content exhibits varying distributions of feature similarity, as shown in Figure 5. We visualize the cosine similarity matrix of features from block index $i \leq 20$ at each denoising step compared to other steps. We find that the distribution of similar features is related to the structural complexity of the generated content. In Figure 5, as structural complexity increases from left to right, the number of similar features suitable for reuse decreases. To further enhance the performance of the BlockDance strategy, we propose a lightweight framework, BlockDance-Ada, designed to learn instance-specific cache and reuse strategies.

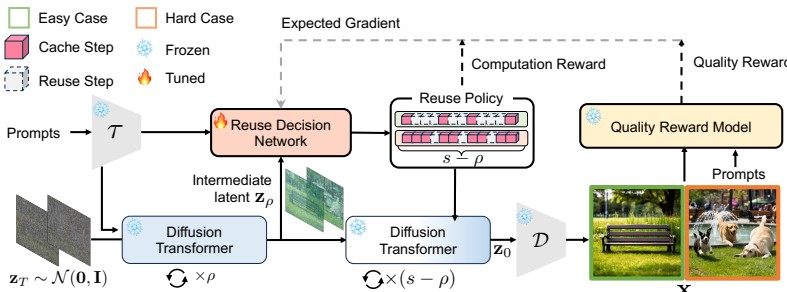

Figure 6: **An overview of BlockDance-Ada.** Given the intermediate latent and prompt embedding, the reuse decision network learns the structural complexity of each sample and derives the corresponding reuse policy. These policies determine whether each subsequent step in DiTs is a cache step or a reuse step. The reward function balances the trade-off between image quality and speed.

BlockDance-Ada leverages reinforcement learning to handle non-differentiable binary decisions, *i.e.* sequential gating, to dynamically determine whether each step in the denoising process is a cache step or a reuse step. As illustrated in Figure 6, for a total of $s$ denoising steps, given latent $\mathbf{z}_T \sim \mathcal{N}(\mathbf{0}, \mathbf{I})$ and text embedding $\mathbf{c} = \tau(\mathbf{p})$ of the prompt $\mathbf{p}$, we initially perform $\rho$ steps of normal computation,*i.e.* cache steps to obtain intermediate latents $\mathbf{z}_\rho$. The state space is defined as $\mathbf{z}_\rho$ and $\mathbf{c}$, and the actions within the decision model are defined to determine whether each step in the subsequent $s - \rho$ steps should be a cache or a reuse step. More formally, the decision network $f_d$, parameterized by $\mathbf{w}$, learns the distribution of feature similarity, then maps it to vectors $\mathbf{m} \in \mathbb{R}^{(s-\rho)}$ :

$$\mathbf{m} = \text{sigmoid}(f_d(\mathbf{z}_\rho, \mathbf{c}; \mathbf{w})). \tag{3}$$

Here, each entry in $\mathbf{m}$ is normalized to be in the range [0, 1], indicating the likelihood of performing a cache step. We define a reuse policy $\pi^f(\mathbf{u} \mid \mathbf{z}_\rho, \mathbf{c})$ with an $(s - \rho)$-dimensional Bernoulli distribution:

$$\pi^f(\mathbf{u}|\mathbf{z}_\rho, \mathbf{c}) = \prod_{t=1}^{s-\rho} \mathbf{m}_t^{\mathbf{u}_t} (1 - \mathbf{m}_t)^{1-\mathbf{u}_t}, \tag{4}$$

where $\mathbf{u} \in \{0, 1\}^{(s-\rho)}$ are actions based on $\mathbf{m}$, and $\mathbf{u}_t = 1$ indicates the $t$-th step is a cache step, and zero entries in $\mathbf{u}$ are reuse steps. During training, $\mathbf{u}$ is produced by sampling from the corresponding

policy, and a greedy approach is used at test time. With this approach, DiTs generate the latent $\mathbf{z}_0$ based on the reuse policy, following the decoder $D$ decodes the latent into a pixel-level image $\mathbf{x}$.

Based on this, we design a reward function to incentivize $f_d$ to maximize computational savings while maintaining quality. The reward function consists of two main components: an image quality reward and a computation reward, balancing generation quality and inference speed. For the image quality reward $\mathcal{Q}(\mathbf{u})$, we use the quality reward model (Xu et al., 2023) $f_q$ to score the generated images based on visual aesthetics and prompt adherence, $i.e.$ $\mathcal{Q}(\mathbf{u}) = f_q(\mathbf{x})$. The computation reward $\mathcal{C}(\mathbf{u})$ is defined as the normalized number of reuse steps, given by the formula:

$$\mathcal{C}(\mathbf{u}) = 1 - \frac{(\sum_{t=1}^{s-\rho} \mathbf{u}_t)}{(s-\rho)}. \tag{5}$$

Finally, the overall reward function is formalized as $\mathcal{R}(\mathbf{u}) = \mathcal{C}(\mathbf{u}) + \lambda \mathcal{Q}(\mathbf{u})$, where $\lambda$ modulates the importance of image quality. At this point, the decision network $f_d$ can be optimized to maximize the expected reward: $\max_{\mathbf{w}} \mathcal{L} = \mathbb{E}_{\mathbf{u} \sim \pi^f} \mathcal{R}(\mathbf{u})$. We use policy gradient methods (Sutton & Barto, 2018) to learn the parameters $\mathbf{w}$ for $f_q$ and the expected gradient can be derived as:

$$\nabla_{\mathbf{w}} \mathcal{L} = \mathbb{E} \left[ \mathcal{R}(\mathbf{u}) \nabla_{\mathbf{w}} \log \pi^f (\mathbf{u} \mid \mathbf{z}_\rho, \mathbf{c}) \right]. \tag{6}$$

We use samples in mini-batches to compute the expected gradient and approximate Eqn. 6 to:

$$\nabla_{\mathbf{w}} \mathcal{L} \approx \frac{1}{B} \sum_{i=1}^{B} \left[ R(\mathbf{u}_i) \nabla_{\mathbf{w}} \log \pi^f (\mathbf{u}_i \mid \mathbf{z}_{\rho_i}, \mathbf{c}_i) \right], \tag{7}$$

where B is the number of samples in the mini-batch. The gradient is then propagated back to train the $f_d$ with Adam (Kingma & Ba, 2014) optimizer. Following this training process, the decision network perceives instance-specific cache and reuse strategies, thereby achieving efficient dynamic inference.

## 4 EXPERIMENTS

### 4.1 EXPERIMENTAL DETAILS

#### 4.1.1 MODELS, DATASETS AND EVALUATION METRICS

To demonstrate the effectiveness of our approach across various generative tasks and types of DiTs, we conduct evaluations on class-conditional image generation, text-to-image generation, and text-to-video generation. For class-conditional image generation, we use DiT-XL/2 (Peebles & Xie, 2023) to generate 50 $512 \times 512$ images per class on the ImageNet (Deng et al., 2009) dataset via DDIM sampler (Song et al., 2021), with a guidance scale of 4.0. For text-to-image generation, we used PixArt-$\alpha$ (Chen et al., 2024a) to generate $1024 \times 1024$ images on the 25K validation set of COCO2017 (Lin et al., 2014) via DPMSolver sampler (Lu et al., 2022), with a guidance scale of 4.5. For text-to-video generation, we used Open-Sora 1.0 (Zheng, 2024) to generate 16-frame videos at $512 \times 512$ resolution on the 2990 test set of MSR-VTT (Xu et al., 2016) via DDIM sampler, with a 7.0 guidance scale. We follow the previous works (Peebles & Xie, 2023; Chen et al., 2024a; Zheng, 2024) to evaluate these tasks and additionally report IQS score (Xu et al., 2023) and Pickscore (Kirstain et al., 2023) for text-to-image generation. We measure inference speed on the A100 GPU by the time it takes the model to generate each image or video, $i.e.$ latency.

#### 4.1.2 IMPLEMENTATION DETAILS

For BlockDance, the cache and reuse steps in PixArt-$\alpha$ are primarily between 40% and 95% of the denoising process, while in DiT-XL/2 and Open-Sora, they are mainly between 25% and 95% of the denoising process. The sizes of the cached features for generating each content with these three models are 18MB, 4.5MB, and 72MB, respectively. The default block index $i$ is set to 20. For BlockDance-Ada, we design the decision network as a lightweight architecture consisting of three transformer blocks and a multi-layer perceptron. The parameters of the decision network amount to 0.08B. We set $\rho$ to 40% of the total number of denoising steps. The parameter $\lambda$ in the reward function is set to 2. For PixArt-$\alpha$, we train the step selection network on 10,000 SAM-LLaVA Captions from its training dataset for 100 epochs with a batch size of 16. We use Adam with a learning rate of $10^{-5}$.

Table 1: Text-to-image generation on PixArt-$\alpha$.

| COCO2017 | Speed | | Image Quality | | | | | |
|---|---|---|---|---|---|---|---|---|
| | MACs (T) ↓ | Latency (s) ↓ | FID ↓ | IS ↑ | CLIP ↑ | IR ↑ | Pick ↑ | SSIM ↑ |
| PixArt-$\alpha$, 30 steps | 128.47 | 3.10 | 30.41 | 39.07 | 0.332 | 0.85 | 22.55 | - |
| ToMe (25% ratio) | 119.34 | 2.70 | 174.57 | 11.68 | 0.302 | -0.47 | 22.19 | 0.18 |
| DeepCache (N=2) | 96.36 | 2.24 | 31.57 | 37.44 | 0.331 | 0.76 | 22.31 | 0.60 |
| TGATE (m=15) | 98.41 | 2.30 | 30.82 | 38.50 | 0.331 | 0.77 | 22.42 | 0.55 |
| PixArt-LCM (8 steps) | 17.13 | 0.83 | 31.67 | 37.83 | 0.328 | 0.58 | 22.25 | 0.41 |
| BlockDance (N=2) | 98.21 | 2.31 (↑ 25.4%) | **30.69** | **38.73** | **0.332** | **0.82** | 22.46 | **0.89** |
| BlockDance (N=3) | 88.11 | 2.09 (↑ 32.6%) | 31.34 | 37.74 | 0.331 | 0.77 | 22.34 | 0.83 |
| BlockDance (N=4) | 81.38 | 1.88 (↑ 39.4%) | 33.28 | 36.48 | 0.330 | 0.72 | 22.21 | 0.79 |

## 4.2 MAIN RESULTS

### 4.2.1 EXPERIMENTS ON THE TRAINING-FREE PARADIGM BLOCKDANCE

**Accelerate PixArt-$\alpha$ for text-to-image generation.** The results on the 25k COCO2017 validation set, as shown in Table 1, demonstrate the efficacy of BlockDance. We extend ToMe (Bolya & Hoffman, 2023) and DeepCache (Ma et al., 2024b) to PixArt-$\alpha$ as baselines. For ToMe, we reduce the computational cost by removing 25% of the tokens through the merge operation. For DeepCache, we reuse features at intervals of 2 throughout the denoising process, specifically reusing the outputs from the first 14 blocks out of the 28 blocks in PixArt-$\alpha$. With $N = 2$, BlockDance accelerates PixArt-$\alpha$ by 25.4% with no significant degradation in image quality, both in terms of visual aesthetics and prompt following. Different speed-quality trade-offs can be modulated by $N$.

Compared to ToMe, BlockDance consistently outperforms ToMe by a clear margin regardless of the reuse frequency $N$. This can be attributed to DiTs featuring a more attention-intensive architecture than the U-Net-based one, thus the continuous use of token merging in DiTs exacerbates quality degradation. Compared to DeepCache, BlockDance achieves better performance across all image metrics at comparable speeds by focusing on high-similarity features. We specifically reduce redundant structural computation in the later stages of denoising, avoiding dissimilar feature reuse and minimizing image quality loss. However, DeepCache reuses features throughout the entire denoising process and does not specifically aim at highly similar features for reuse. This leads to the inclusion of dissimilar features in the reused set, resulting in structural distortions and a decline in prompt alignment. Compared to TGATE (Zhang et al., 2024b), which accelerates by reducing the redundancy in cross-attention calculations. Blockdance supports DiTs that do not incorporate cross-attention, such as SD3 (Esser et al., 2024) and Flux (Labs, 2024). Besides, experimental results show that with the same acceleration benefit, BlockDance outperforms TGATE across various metrics. Compared to PixArt-LCM (Chen et al., 2024b) obtained through consistency distillation training, BlockDance, although requiring more inference time, achieves higher generation quality across multiple metrics without additional training. It is worth noting that BlockDance's generated images exhibit higher consistency with the base model, as evidenced by significantly better SSIM performance compared to baselines, thanks to our targeted reuse strategy.

Table 2: Class-conditional generation on DiT-XL/2 via 50 DDIM steps.

| ImageNet | Speed | | Image Quality | | | |
|---|---|---|---|---|---|---|
| | MACs (T) ↓ | Latency (s) ↓ | FID ↓ | sFID ↓ | IS ↑ | SSIM ↑ |
| DiT-XL/2 | 45.71 | 1.79 | 15.89 | 21.01 | 413.8 | - |
| ToMe (25% ratio) | 41.90 | 1.61 (↑ 10.1%) | 27.24 | 53.46 | 176.01 | 0.44 |
| DeepCache (N=2) | 34.28 | 1.10 (↑ 38.5%) | 16.11 | 28.18 | 392.29 | 0.90 |
| BlockDance (N=2) | 29.91 | 1.12 (↑ 37.4%) | **15.70** | **22.86** | **402.01** | **0.98** |
| BlockDance (N=3) | 24.16 | 0.90 (↑ 49.7%) | 15.96 | 24.43 | 390.83 | 0.95 |
| BlockDance (N=4) | 19.85 | 0.76 (↑ 57.5%) | 16.01 | 25.28 | 383.61 | 0.93 |

Table 3: Text-to-video generation on Open-Sora. All the methods here adopt 100 DDIM steps.

| MSR-VTT | Speed | | Video Quality | | | |
|---|---|---|---|---|---|---|
| | MACs (T) ↓ | Latency (s) ↓ | FVD ↓ | KVD ↓ | CLIP ↑ | IS ↑ |
| Open-Sora | 2193.80 | 44.99 | 548.72 | 74.03 | 0.299 | 20.27 |
| DeepCache (N=2) | 1644.75 | 27.58 (↑ 38.7%) | 942.06 | 108.4 | 0.298 | 18.20 |
| BlockDance (N=2) | 1418.28 | 29.32 (↑ 34.8%) | **550.22** | **72.35** | **0.299** | **20.22** |
| BlockDance (N=3) | 1159.76 | 24.66 (↑ 45.2%) | 580.35 | 73.70 | 0.297 | 19.92 |
| BlockDance (N=4) | 970.18 | 20.39 (↑ 54.7%) | 674.26 | 86.82 | 0.291 | 18.81 |

**Accelerate DiT/XL-2 for class-conditional generation.** The results of the 50k ImageNet images are shown in Table 2. We extend ToMe and DeepCache to DiT/XL-2 as the baselines. At $N = 2$, BlockDance not only accelerates DiT/XL-2 by 37.4% while maintaining image quality but also outperforms DeepCache while preserving higher consistency with the base model's generated images. By increasing, the acceleration ratio of BlockDance can reach up to 57.5%, and the image quality consistently outperforms ToMe.

**Accelerate Open-Sora for text-to-video generation.** BlockDance is also effective for accelerating video generation tasks. The accelerated results on MSR-VTT are shown in Table 3. At $N = 2$, BlockDance speeds up Open-Sora by 34.8% while maintaining video quality, both in terms of visual quality and temporal consistency. Increasing $N$ achieves various quality-speed trade-offs. In contrast, DeepCache suffers from significant quality degradation, as evidenced by the deterioration in metrics such as FVD. This is attributed to DeepCache reusing low-similarity features, such as structural information at the early stages of denoising.

Figure 7: Qualitative Results. Compared to previous methods, BlockDance achieves not only high fidelity but also a high degree of consistency with the original images.

**Qualitative results.** We further qualitatively analyze our approach as shown in Figure 7. ToMe merges adjacent similar tokens to save self-attention computation, but this method is not very friendly for transformer-intensive architectures, resulting in low-quality images with "blocky artifacts". While DeepCache and TGATE achieve approximately 27% acceleration, they may cause significant structural differences from the original images and present artifacts and semantic loss in some complex cases. PixArt-LCM accelerates PixArt-$\alpha$ through additional consistency distillation training, yielding significant acceleration but with noticeable declines in visual aesthetics and prompt following. In contrast, BlockDance achieves a 25.4% acceleration without additional training costs, while maintaining high consistency with the original model in terms of structure and detail.

### 4.2.2 EVALUATION ON BLOCKDANCE-ADA

Table 4: BlockDance-Ada achieves a better trade-off between quality and speed by dynamic inference.

| COCO 2017 | Latency (s) ↓ | FID ↓ | IS ↑ | CLIP ↑ | IR ↑ | Pick ↑ |
|---|---|---|---|---|---|---|
| PixArt-$\alpha$, 30steps | 3.10 | 30.41 | 39.07 | 0.332 | 0.85 | 22.55 |
| ToMe (25% ratio) | 2.70 | 174.57 | 11.68 | 0.302 | -0.47 | 22.19 |
| DeepCache (N=2) | 2.24 | 31.57 | 37.44 | 0.331 | 0.76 | 22.31 |
| TGATE (m=15) | 2.30 | 30.82 | 38.50 | 0.331 | 0.77 | 22.42 |
| BlockDance (N=2) | 2.31 (↑ 25.4%) | 30.69 | 38.73 | 0.332 | 0.82 | 22.46 |
| BlockDance (N=3) | 2.09 (↑ 32.6%) | 31.34 | 37.74 | 0.331 | 0.77 | 22.34 |
| BlockDance-Ada | 2.15 (↑ 30.6%) | 30.71 | 38.70 | 0.332 | 0.81 | 22.44 |

**Dynamic Inference on PixArt-$\alpha$.** Table 4 provides a detailed breakdown of the performance of BlockDance-Ada. By dynamically allocating computational resources for each sample based on

instance-specific strategies, BlockDance-Ada effectively reduces redundant computation, achieving acceleration close to that of BlockDance ($N = 3$) while delivering superior image quality. Compared to BlockDance ($N = 2$), BlockDance-Ada offers greater acceleration benefits with similar quality.

## 4.3 DISCUSSION.

### 4.3.1 ABLATION STUDY.

**Impact of reuse frequency.** As shown in Figure 8, we illustrate how the generated images evolve as $N$ increases. With the reduction in generation time, the main subject of the image remains consistent, but the fidelity of the details gradually decreases, aligning with the insights presented in Table 1. Different values of $N$ offer flexible choices for various speed-quality trade-offs.

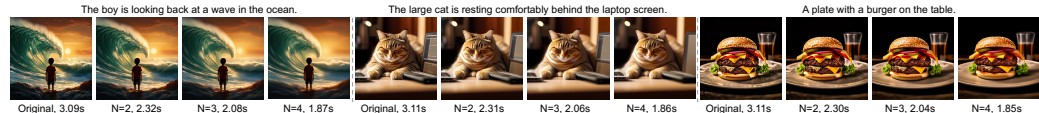

Figure 8: Ablation study on the effect of reuse frequency on generated images.

**Impact of applying BlockDance in different denoising stages.** As shown in Figure 9, we investigate the impact of applying BlockDance in different stages of the denoising process: the initial stage (0%-40%) and the later stage (40%-95%). BlockDance primarily reuses structural features; therefore, applying it at the initial stage that focuses on the structure may result in structural changes or artifacts (highlighted by the red box in Figure 9 (a)), as the structure has not yet stabilized. Conversely, in the later stage, where structural information has stabilized and the focus shifts towards texture details, reusing structural features accelerates inference with minimal quality loss, as shown in Figure 9 (b).

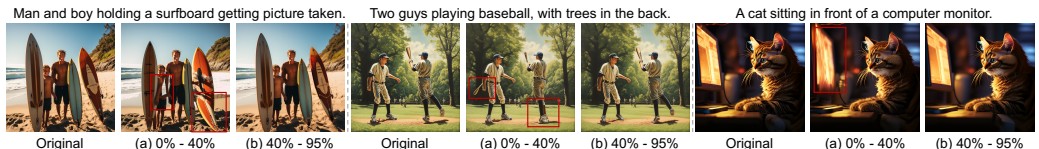

Figure 9: Effect of using BlockDance at different denoising stages.

**Impact of reusing blocks at different depths on generated images.** We investigate the impact of reusing only the shallow and middle blocks versus reusing deeper blocks as well in the transformer, as shown in Figure 10. Due to the low similarity of features in the deeper blocks, reusing them results in the loss of computation related to details, leading to degradation in texture details, as highlighted by the red boxes in Figure 10 (b). Conversely, reusing the higher similarity shallow and middle blocks, which focus on structural information, results in minimal quality degradation.

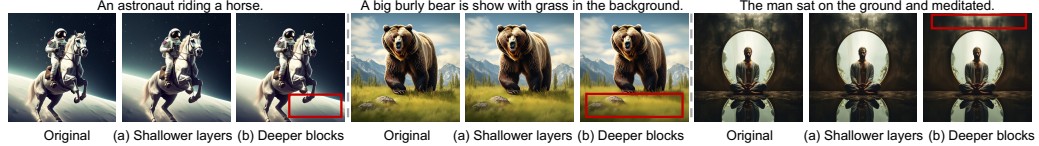

Figure 10: Impact of reusing blocks at different depths on generated images.

## 5 CONCLUSION

In this paper, we introduce BlockDance, a novel training-free acceleration approach for DiTs that leverages the redundancy across adjacent denoising steps. During the denoising process, by caching and reusing structure-level features after the structure has stabilized, *i.e.*, structurally similar spatio-temporal features, BlockDance significantly accelerates DiTs with minimal quality loss and maintains high consistency with the base model. Additionally, we propose BlockDance-Ada, a variant of BlockDance that dynamically allocates computational resources based on instance-specific reuse policies, further enhancing the efficiency of DiTs inference while maintaining superior image quality.

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

# A  APPENDIX

## A.1  ADDITIONAL EXPERIMENTS

**Accelerate at any number of steps.**  The acceleration paradigm we proposed is complementary to other acceleration techniques and can be used on top of them for further enhancement. Here, we validate the performance of BlockDance across different sampling steps for each model. As demonstrated in Tables 5, 6, and 7, BlockDance effectively accelerates the process across various step counts while maintaining the quality of the generated content.

Table 5: Accelerate PixArt-$\alpha$ at any number of steps. All the methods adopt the DPM-Solver sampler.

| COCO2017 | PixArt-$\alpha$ | | BlockDance (N=2) | |
|---|---|---|---|---|
| | Latency ↓ | FID ↓ | Latency ↓ | FID ↓ |
| step=20 | 2.02 | 30.79 | 1.51 (↑ 24.8%) | 30.87 |
| step=30 | 3.10 | 30.41 | 2.31 (↑ 25.4%) | 30.69 |
| step=40 | 4.15 | 30.19 | 3.08 (↑ 25.8%) | 30.35 |

Table 6: Accelerate DiT-XL/2 at any number of steps. All the methods here adopt the DDIM sampler.

| ImageNet | DIT-XL/2 | | BlockDance (N=2) | |
|---|---|---|---|---|
| | Latency ↓ | FID ↓ | Latency ↓ | FID ↓ |
| step=30 | 1.07 | 16.15 | 0.67 (↑ 37.3%) | 16.06 |
| step=40 | 1.43 | 16.04 | 0.90 (↑ 37.1%) | 15.91 |
| step=50 | 1.79 | 15.89 | 1.12 (↑ 37.4%) | 15.70 |

Table 7: Accelerate Open-Sora at any number of steps. All the methods here adopt the DDIM sampler.

| MSR-VTT | Open-Sora | | BlockDance (N=2) | |
|---|---|---|---|---|
| | Latency ↓ | FVD ↓ | Latency ↓ | FVD ↓ |
| step=50 | 27.72 | 582.91 | 18.16 (↑ 34.5%) | 585.21 |
| step=75 | 36.53 | 561.72 | 23.78 (↑ 34.9%) | 562.83 |
| step=100 | 44.99 | 548.72 | 29.32 (↑ 34.7%) | 550.22 |

**Accelerate SD3 for text-to-image generation.**  To validate the effectiveness of our proposed paradigm across different DiT architecture variants, we apply BlockDance to MMDiT-based DiT models (Esser et al., 2024; Labs, 2024), such as Stable Diffusion 3 (Esser et al., 2024). The results are conducted on the 25k COCO2017 validation set, as shown in Table 8. The experimental results indicate that with $N = 2$, BlockDance accelerates SD3 by 25.3% while maintaining comparable image quality, both in terms of visual aesthetics and prompt following. Different speed-quality trade-offs can be modulated by N.

**Quantitative results of the ablation on PixArt-$\alpha$.**  The quantitative results of the ablation experiments on the impact of applying BlockDance at different denoising stages are shown in Table 9. These results are consistent with the conclusions drawn in Figure 9, indicating that reducing redundant computation related to structural information after the structure has stabilized can accelerate inference with minimal quality loss. The quantitative results of the ablation experiments on the impact of reusing blocks at different depths on generated images are shown in Table 10. These results align with the conclusions in Figure 10, showing that reusing only the structure-focused blocks, *i.e.* shallow and middle blocks, leads to better image quality.

Table 8: Text-to-image generation on SD3.

| Model | Latency (s) ↓ | IR ↑ | Pick ↑ | IS ↑ | CLIP ↑ | FID ↓ | SSIM ↑ |
|---|---|---|---|---|---|---|---|
| SD3 | 4.35 | 1.01 | 22.49 | 41.52 | 0.334 | 26.95 | - |
| BlockDance(N=2) | 3.25 (↑25.3%) | **1.00** | **22.45** | **40.89** | **0.334** | **27.52** | **0.96** |
| BlockDance(N=3) | 2.99 (↑31.3%) | 0.99 | 22.42 | 40.52 | 0.334 | 27.74 | 0.95 |
| BlockDance(N=4) | 2.74 (↑37.0%) | 0.98 | 22.34 | 39.53 | 0.334 | 28.42 | 0.92 |

Table 9: Ablation on denoising stage.

| Model | Latency (s) ↓ | IR ↑ | Pick ↑ | IS ↑ | SSIM ↑ |
|---|---|---|---|---|---|
| PixArt-$\alpha$ | 3.1 | 0.85 | 22.55 | 39.07 | - |
| BlockDance, 0%∼40% | 2.46 (↑ 20.6%) | 0.76 | 22.31 | 37.69 | 0.79 |
| BlockDance, 40%∼95% | 2.31 (↑ 25.4%) | 0.82 | 22.46 | 38.73 | 0.89 |

Table 10: Ablation on reusing at different blocks.

| Model | Latency (s) ↓ | IR ↑ | Pick ↑ | IS ↑ | SSIM ↑ |
|---|---|---|---|---|---|
| PixArt-$\alpha$ | 3.1 | 0.85 | 22.55 | 39.07 | - |
| BlockDance, Deep | 2.22 (↑ 28.4%) | 0.79 | 22.39 | 38.24 | 0.85 |
| BlockDance, Shallow | 2.31 (↑ 25.4%) | 0.82 | 22.46 | 38.73 | 0.89 |

**More qualitative results.** To comprehensively verify the method we proposed, we present additional qualitative results for each DiT model, as indicated in Figures 11, 12, and 13. Our method maintains high-quality content with a high degree of consistency with the content generated by the original models, while achieving significant acceleration.

**Limitation.** Although BlockDance accelerates various DiT models across various generative tasks in a plug-and-play manner, its application is limited in scenarios with very few denoising steps (*e.g.*, 1 to 4 steps), due to the reduced similarity of features between adjacent steps. However, in scenarios where most base models use a larger number of steps during inference, training a distilled version with fewer steps for each base model incurs high training costs and time consumption, whereas our method requires no additional training costs and operates in a plug-and-play manner.

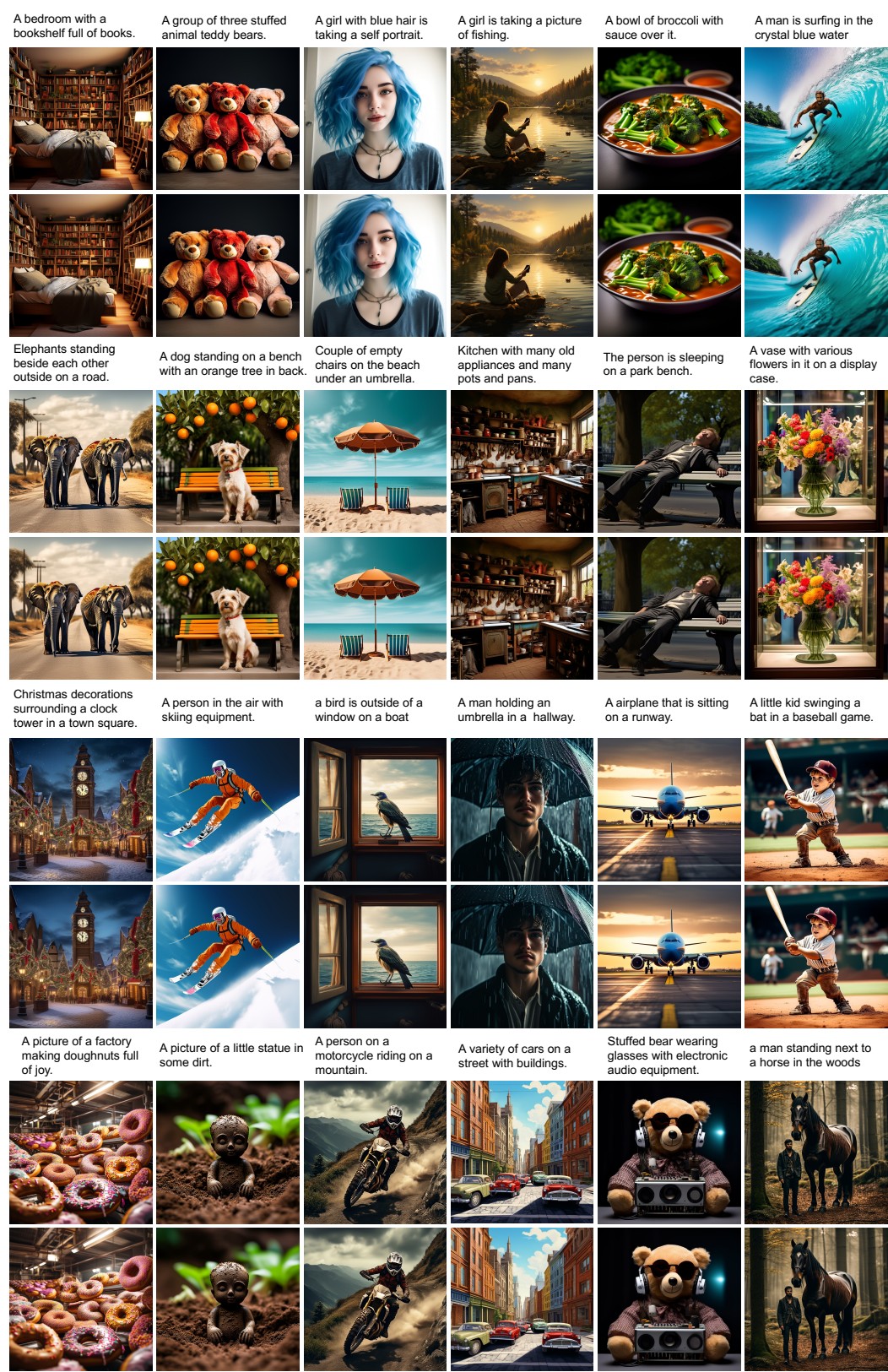

Figure 11: PixaArt-$\alpha$: Samples with 30 DPM-Solver steps (upper row) and 30 DPM-Solver steps + BlockDance with $N = 2$ (lower row). Our method speeds up 25.4% while maintaining the visual aesthetics and prompt following. Here, prompts are selected from the COCO2017 validation set.

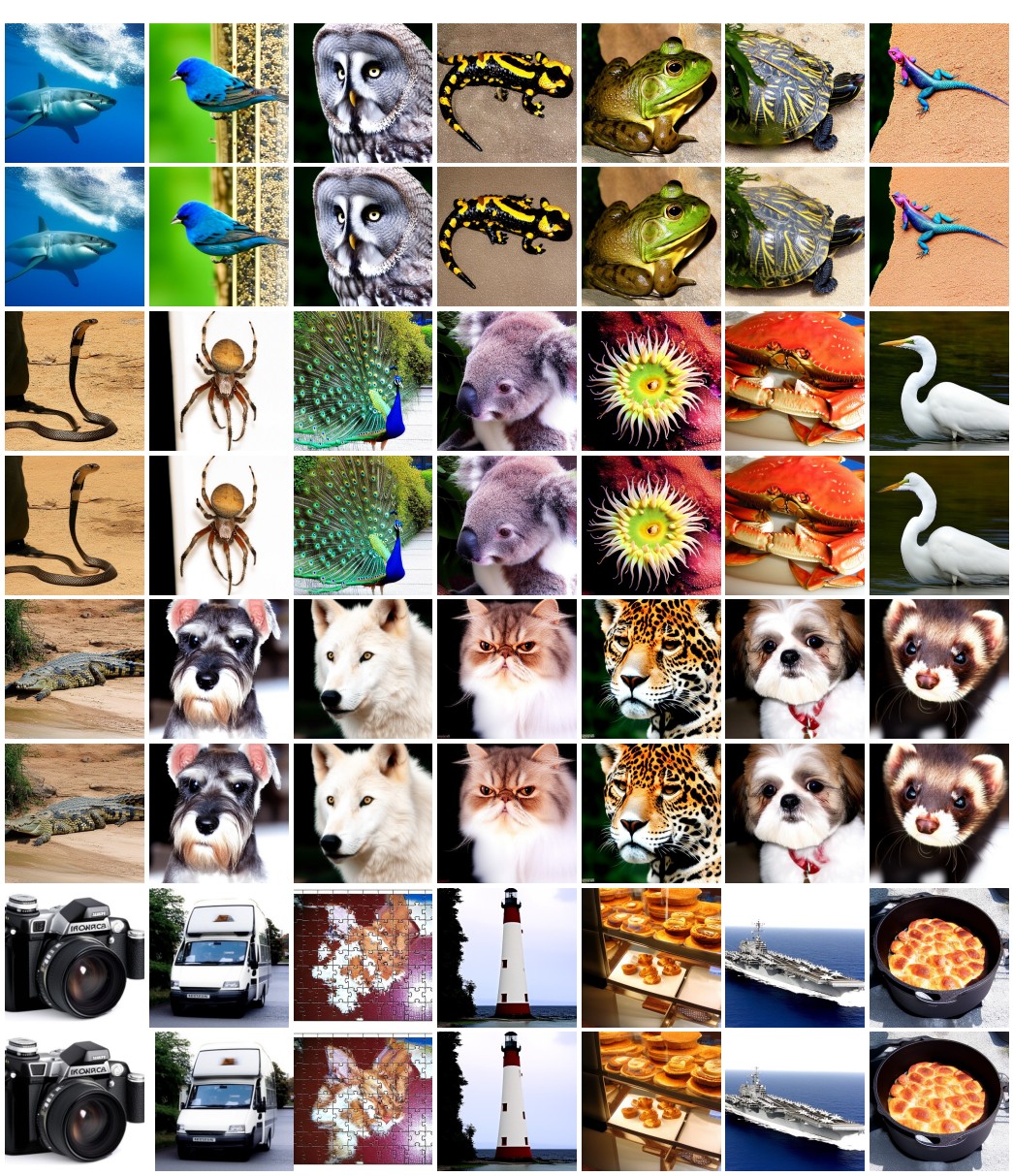

Figure 12: DiT-XL/2 for ImageNet: Samples with 50 DDIM steps (upper row) and 50 DDIM steps + BlockDance with $N = 2$ (lower row). Our method achieves a 37.4% acceleration while maintaining image quality.

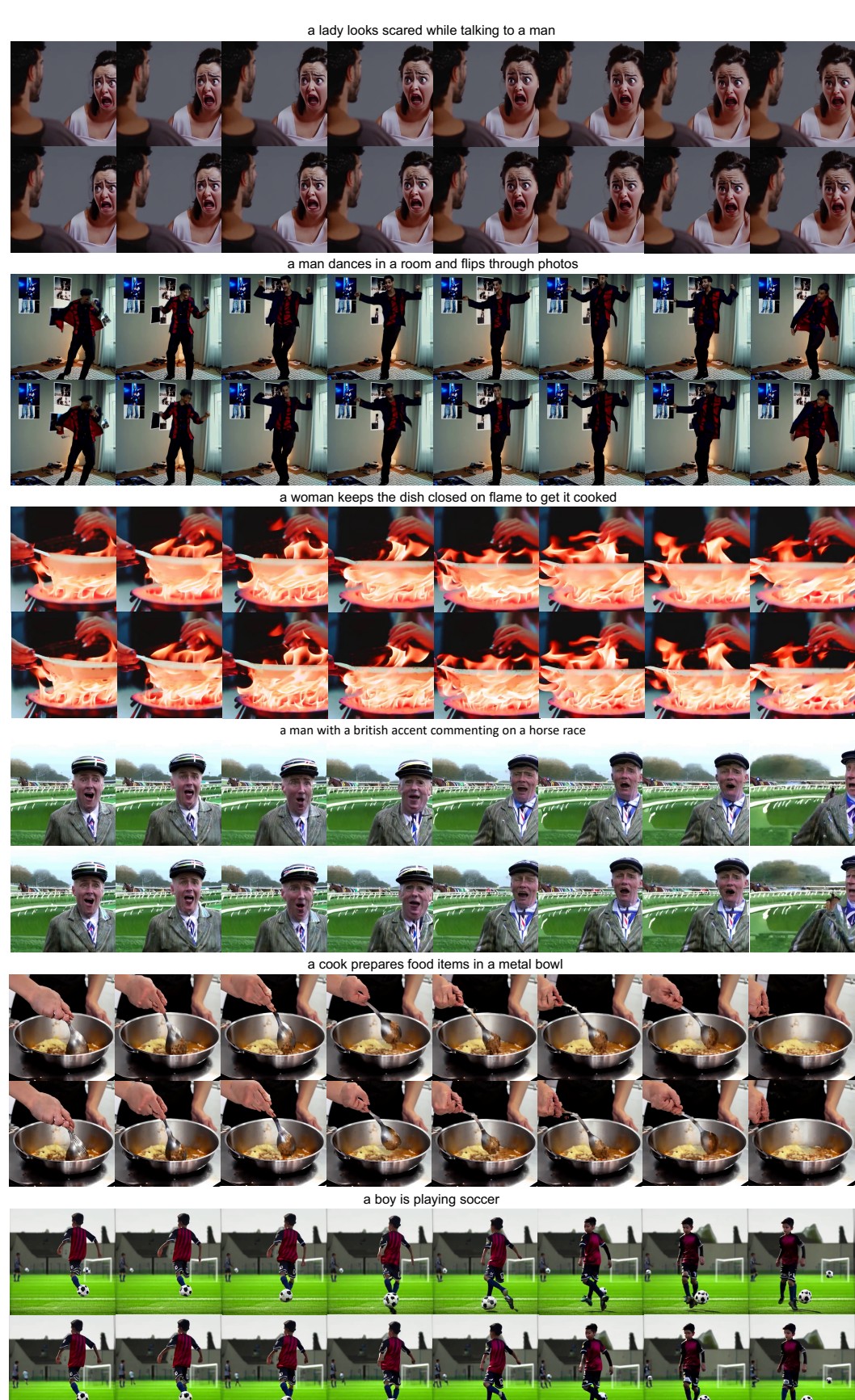

Figure 13: Open-Sora: Samples with 100 DDIM steps (upper row) and 100 DDIM steps + BlockDance with $N = 2$ (lower row). Our method achieves a 34.8% acceleration while maintaining visual quality and high motion consistency with the original video.

