# OpenReview forum: "BlockDance: Reuse Structurally Similar Spatio-Temporal Features to Accelerate Diffusion Transformers"
_ICLR.cc/2025/Conference — ICLR 2025 Conference Withdrawn Submission_

### Official Review · Reviewer_CUFA · 2024-10-31

**Soundness:** 2
**Presentation:** 2
**Contribution:** 2
**Rating:** 5
**Confidence:** 5

**Summary:**

This work proposes a training-free acceleration method for DiTs based on caching, named BlockDance. It caches and uses similar features during the later stages of denoising to reduce computation. BlockDance-Ada further uses a lightweight network to learn instance-specific acceleration strategies. They achieve 25-50% acceleration.

**Strengths:**

1. easy to follow

2. BlockDance-Ada can significantly recover the performance drops of BlockDance.

3. The experiments include DiT, PixArt, and open-sora, making the evaluation comprehensive and convincing.

**Weaknesses:**

1. The proposed method accelerates DiT-based models by only about 30% while incurring obvious performance degradation, making its practical application questionable especially compared to other techniques like quantization and distillation. Even the BlockDance-Ada version in Table 4 does not show impressive results.

2. BlockDance-Ada leverages reinforcement learning. What's the training cost?

**Questions:**

How was the inference latency evaluated? Are they all performed on A100 GPU? How is it affected by parameters like batch size?

---

### Official Review · Reviewer_FEkN · 2024-11-03

**Soundness:** 2
**Presentation:** 3
**Contribution:** 1
**Rating:** 5
**Confidence:** 4

**Summary:**

This paper proposes a method called BlockDance to accelerate the denoising process of DiT. BlockDance caches specific feature maps from the DiT model, allowing them to be reused at the next timestep and thereby skipping computations from previous layers. Additionally, the authors’ proposed BlockDance-Ada enables the use of an optimal caching strategy tailored to each data sample.

**Strengths:**

- This paper is well-written and easy to understand.
- Experiments were conducted on various datasets, including video generation datasets like Open-Sora, not just for image generation.
- The proposed BlockDance-Ada is novel.

**Weaknesses:**

- **Novelty Issue:** Both the proposed feature similarity and caching methods have already been introduced in previous works [1], [2]. The main contribution of this paper appears to be the proposal of BlockDance-Ada. However, the experimental results provided are limited, and the performance improvement is marginal. For instance, in Table 4, BlockDance-Ada shows a 0.15s latency reduction compared to BlockDance (N=2), but with a 0.02 increase in FID. This minor improvement may simply reflect a trade-off between latency and FID.
- **Lack of Comparative Experimental Results:** The paper does not provide sufficient evidence that BlockDance is genuinely faster than existing methods. To accurately compare performance, an evaluation on a Pareto curve with latency and FID as the axes would be more informative.
- **Lack of Persuasiveness in Fig. 5:** Figure 5 is unconvincing. At first glance, it suggests that the diffusion model maintains a high level of similarity across nearly all timesteps.
- **Missing Comparison with Recent Methods:** The paper lacks a comparison with recent caching-based acceleration methods, such as [2] and [3].

[1] Xu, Mengwei, et al. "Deepcache: Principled cache for mobile deep vision." Proceedings of the 24th annual international conference on mobile computing and networking. 2018.

[2] So, Junhyuk, Jungwon Lee, and Eunhyeok Park. "FRDiff: Feature Reuse for Universal Training-free Acceleration of Diffusion Models." arXiv preprint arXiv:2312.03517 (2023).

[3] Li, Senmao, et al. "Faster diffusion: Rethinking the role of unet encoder in diffusion models." arXiv e-prints (2023): arXiv-2312.

**Questions:**

- When training BlockDance-Ada, is it possible to learn not only the timestep but also the caching index? Since each Block in DiT performs different tasks during denoising, this could be very beneficial.
- Can BlockDance be applied to UNet-based Diffusion Models as well, rather than just DiT?

---

### Official Review · Reviewer_dwuN · 2024-11-03

**Soundness:** 3
**Presentation:** 3
**Contribution:** 2
**Rating:** 5
**Confidence:** 5

**Summary:**

BlockDance is introduced as a novel, efficient algorithm designed to accelerate Diffusion Transformers (DiTs) without requiring additional training. By caching and reusing Structurally Similar Spatio-Temporal (STSS) features, BlockDance reduces redundant computations during inference, making it adaptable and compatible with a range of models in a plug-and-play manner. The approach also explores instance-specific optimization through BlockDance-Ada, which uses reinforcement learning to adaptively conserve computation based on the complexity of generated content. Extensive validations on datasets like ImageNet, COCO2017, and MSR-VTT across tasks such as class-conditioned generation, text-to-image, and text-to-video demonstrate BlockDance’s ability to achieve 25%-50% faster inference while maintaining quality, with BlockDance-Ada further enhancing output quality under the same acceleration conditions.

**Strengths:**

1. The BlockDance-Ada component presents a novel approach by enabling an instance-dynamic caching strategy, which intelligently adapts to the varying complexity of generated content. This adaptive mechanism sets it apart from traditional static caching methods and demonstrates the use of reinforcement learning techniques for efficient computation management.

2. The approach has undergone extensive validation across multiple datasets, including ImageNet, COCO2017, and MSR-VTT. These experiments showcase the algorithm’s generalizability and effectiveness in diverse settings.

**Weaknesses:**

1. The paper does not adequately address prior research efforts such as FORA [1], Delta DiT [2], and PAB [3]. The motivation highlighted in Figure 2, which describes the reduction of redundant computations, has been extensively covered in these works. Additionally, these relevant studies are not referenced or compared in the experimental section, limiting the contextual understanding of how BlockDance differentiates or builds on these methodologies.

2. The first contribution appears to have a significant overlap with existing work, suggesting that the novelty is relatively incremental. Without a deeper analysis or clearer distinction from prior caching strategies, the innovation might seem limited compared to the foundational ideas already established.

3. The practical impact of BlockDance seems constrained. As seen in Table 4, the adaptive strategy BlockDance-Ada only achieves about a 5% further reduction in computation compared to the standard BlockDance with 𝑁=2, indicating that the improvements, while present, may not justify the additional complexity introduced.


[1] FORA: Fast-Forward Caching in Diffusion Transformer Acceleration

[2] $\Delta $-DiT: A Training-Free Acceleration Method Tailored for Diffusion Transformers

[3] Real-Time Video Generation with Pyramid Attention Broadcast

**Questions:**

1. The training cost of BlockDance-Ada should be analyzed with learn-to-cache [1]. Learn-to-cache techniques involve a significant training phase to optimize cache utilization, potentially leading to longer overall preparation times before deployment. However, a detailed cost comparison highlighting computation hours, resources needed, and efficiency gains would provide a clearer understanding of the cost benefits of BlockDance-Ada.

2. The extra cost associated with employing a reward model during the inference stage, particularly in the BlockDance-Ada variant, is important to evaluate. While BlockDance-Ada introduces adaptiveness for higher content quality, this incurs a computational overhead.


[1] Learning-to-Cache: Accelerating Diffusion Transformer via Layer Caching

---

### Official Review · Reviewer_Z2BP · 2024-11-04

**Soundness:** 3
**Presentation:** 3
**Contribution:** 2
**Rating:** 5
**Confidence:** 3

**Summary:**

This study revisits the feature correlation in diffusion transformers and proposes a straightforward, training-free strategy to accelerate generation by caching and reusing features across time steps. The proposed strategy demonstrates effectiveness through experiments on several generation architectures.

**Strengths:**

The idea is straightforward and easy to implement, and the paper is well-written.

The intuitive demonstrations illustrate the generation dynamics of the diffusion transformer, supporting the effectiveness of the proposed strategy.

**Weaknesses:**

Although the block-wise optimization approach differs from DeepCache, i.e., the authors arguing that DeepCache does not specifically aim at highly similar features for reuse, the core concept of caching intermediate features for reuse is to some extent similar. The implementation and experimental impact of reusing similar features in this work appear somewhat marginal. See comments below for more details.

Experimental Results: While the proposed method shows slight improvements over DeepCache, the advantage remains limited, as seen in Tables 1, 2, and 4.

As summarized in the Related Works section, many existing approaches aim to reduce the number of sampling steps. Although this study involves dropping specific diffusion blocks within the network, it bears substantial similarity to step-reduction strategies. Comparisons with previous works (e.g., [1]) appear insufficiently comprehensive. Please note [1] is just one of many possible relevant comparisons.

The research first introduces a manually designed caching strategy, BlockDance, which I would like to consider as a vanilla baseline, and subsequently an instance-wise learnable strategy, BlockDance-Ada, for adaptively reusing features. While this adaptive reuse approach seems reasonable and to be a highlight of the paper, its demonstrated improvement seems limited (Table 4).

Minor Comments:

Claimed Conclusion: “Unlike .., BlockDance prioritizes the identification of the most structurally similar features, referred to as Structurally Similar Spatio-Temporal (STSS) features” Since MSE is used to quantify similarity, it is unclear why these features are referred to as the most ‘structurally’ similar. Could the authors clarify how these features represent 'structural' characteristics?

Claimed Demonstration: “while the deeper blocks shift their focus towards generating more complex high-frequency texture information, such as clouds and crowds within depth of field.” This difference is not immediately apparent to me. Could the authors indicate these distinctions with arrows or highlights?

Claimed Summary: “Several studies (Ma et al., 2024b; Li et al., 2023) have unearthed the existence of redundant features in U-Net-based diffusion models, but their coarse-grained feature reuse strategies include those low-similarity features, leading to structural distortions and text-image misalignment.” Could experimental evidence be provided to support this conclusion?

Figure 5: The text font is too small, which could be enhanced for readability.

[1] Li L, Li H, Zheng X, et al. Autodiffusion: Training-free optimization of time steps and architectures for automated diffusion model acceleration, ICCV, 2023.

**Questions:**

Key concerns are listed in the weakness section, which mainly involves the novelty, experiment, and demonstration issues. Addressing these concerns, especially regarding the experimental aspect, is most valuable to change my rating. Some demonstration questions are also summarized above, clarifications for these questions will also be appreciated.

---

### Note · Authors · 2024-11-15

I have read and agree with the venue's withdrawal policy on behalf of myself and my co-authors.